# The Early Northward Migration of the White-Backed Planthopper (*Sogatella furcifera*) is Often Hindered by Heavy Precipitation in Southern China during the Preflood Season in May and June

**DOI:** 10.3390/insects10060158

**Published:** 2019-06-04

**Authors:** Hui Chen, Xiao-Li Chang, Yun-Ping Wang, Ming-Hong Lu, Wan-Cai Liu, Bao-Ping Zhai, Gao Hu

**Affiliations:** 1College of Plant Protection, Nanjing Agricultural University, Nanjing 210095, China; 2016102100@njau.edu.cn (H.C.); 2017102066@njau.edu.cn (Y.-P.W.); bpzhai@njau.edu.cn (B.-P.Z.); 2Shanghai Academy of Agricultural Science, Shanghai 201403, China; xlchang981@126.com; 3Division of Pest Forecasting, China National Agro-Tech Extension and Service Center, Beijing 100026, China; luminghong@agri.gov.cn (M.-H.L.); liuwancai@agri.gov.cn (W.-C.L.)

**Keywords:** *Sogatella furcifera*, Western Pacific Subtropical High-Pressure, insect migration, seasonal atmospheric circulation

## Abstract

Seasonal weather systems that establish prevailing winds and seasonal rainfall on a large scale largely determine insect migration patterns, especially for micro-insects with completely windborne migration. Recent studies indicated that the summer migration of the brown planthopper (BPH, *Nilaparvata lugens*) in eastern China is related to the strength and position of the Western Pacific Subtropical High-Pressure (WPSH) system and its associated wind and rainfall patterns. Compared with the BPH, the white-backed planthopper (WBPH, *Sogatella furcifera*) has a similar diet, analogous body size, and strong long-distance migration ability. Thus, the migration pattern for the WBPH can be speculated to be similar to that of the BPH. However, the migration pattern of the WBPH and how this pattern relates to climatic conditions have scarcely been described. Based on almost three decades of data (1977–2003), it was suggested that the WBPH in southern China (south of approximately 27° N) migrates into the middle and lower reaches of the Yangtze River after the abrupt movements of the WPSH in mid-June, similar to the BPH. By contrast, the emigration of the WBPH in southern China begins in late May. Further analysis indicated that the migration of the WBPH in late May and early June was short or unsuccessful due to heavy precipitation during the preflood season in southern China from late May to middle June. The results herein demonstrate the differences in migration patterns between two rice planthoppers in the eastern Asia migration arena. We also provide new information that could assist with forecasting outbreaks and implementing control measures against these migratory pests.

## 1. Introduction

Responding to seasonal changes in habitats and resources, trillions of insects migrate through the atmosphere every year [1]. These long-distance movements transport vast quantities of energy, nutrients, pathogens, and parasites, while the migrants themselves often provide ecosystem services (pest control, pollination) or disservices (crop pests, disease vectors) [1,2,3,4]. Generally, flying insects have a small body with limited flight ability; thus, they take the advantage of high-speed airstreams to perform their seasonal migration and their journey can be up to hundreds of or thousands of kilometers [4,5]. In addition, migrating individuals terminate their migration in some weather events, such as rain, strong convergence, and cold temperatures, resulting in a large number of individuals concentrated on the ground in either favorable or unfavorable conditions [6]. Therefore, the long-distance migration of insects is greatly affected by meteorological factors and weather systems, and the extraordinary adaptations of insects for dealing with atmospheric conditions during their migratory flights have captivated entomologists for decades [7].

Seasonal weather systems that establish prevailing winds and seasonal rainfall over thousands of square kilometers, or even large portions of the earth for days or weeks, largely control insect migration patterns, especially for micro-insects with completely windborne migration [7,8,9]. As synoptic wind patterns shape the insect migration pathway and rain zones terminate insect migration, annual variation in seasonal weather systems regulates the timing and intensity of insect migration events and determines the abundance of the subsequent population build-up [3,8,9]. Understanding the relationship between seasonal weather systems and insect migration is therefore of great importance.

The brown planthopper (BPH, *Nilaparvata lugens*) and the white-backed planthopper (WBPH, *Sogatella furcifera*), collectively known as rice planthoppers (RPHs), have periodically erupted across Asia since the 1960s. Outbreaks can result in heavy rice loss and almost total crop failure in many paddies [10,11,12,13]. Neither RPH species can overwinter in the temperate regions of China, Korea, and Japan and outbreaks in these regions each summer are initiated by several waves of windborne spring or summer migrants originating in winter-breeding areas in Indochina [11,14,15,16]. Recently, the interaction of BPH migration and synoptic weather patterns driven by the semi-permanent Western Pacific Subtropical High-Pressure (WPSH) system was well described in our previous works [8,9]. The WPSH is one of the most important atmospheric circulations that influences the weather and climate in eastern Asia [17,18]. The seasonal movement of the WPSH consists of a northward advance and a southward retreat that controls the annual position of rain belts in eastern Asia [17,18]. In spring, the WPSH moves gradually toward the north, which produces several continuous precipitation events in southern China in April and May. This period is called the preflood season in southern China. During the summer season, the WPSH hops abruptly northwards in two steps. This ‘first abrupt jump’ occurs in mid-June when the WPSH expands northward abruptly from southern China to the Yangtze River basin, indicating the Meiyu season in the latter region (and much further afield in Korea and southern Japan) [17,18]. The BPH aerial transport and concentration processes are related to the strength and position of the WPSH and its associated wind and rainfall patterns [8,9]. Major zonal rainfall belts and their associated downdrafts, rain, and cold temperatures form a barrier to BPH flight and promote concentration and landing (e.g., Crummay and Atkinson 1997; Hu et al. 2007, 2019) [8,19,20]. The development of strong southwesterly winds, particularly the development of nocturnal low-level jets, provides rapid aerial transport (e.g., Hu et al., 2007, 2019) [8,20]. In particular, BPHs from southern China (south of approximately 27° N) migrate into the Lower Yangtze River in June and July and synchronize with the ‘first abrupt jump’ of the WPSH [8,9]. Using BPH abundance in source areas in the northern part of southern China (approximately 23.5–27° N, green oval area in Figure 1) and WPSH intensity or related sea surface temperature anomalies, some models were developed to predict planthopper numbers migrating into the key rice-growing area of the Lower Yangtze Valley [8]. Compared with the BPH, the WBPH has a similar diet, analogous body sizes, and a strong long-distance migration ability; thus, the migration pattern of WBPHs can be speculated to be similar to that of BPHs.

The body size of BPHs and WBPHs is similar, and both species also have similar population dynamics in the central Indochina peninsula [16]. Nevertheless, the WBPH first moved into southern China in larger numbers, compared with the BPH under the same weather conditions, and thus, the WBPH established populations earlier than the BPH in southern China [16]. Subsequently, the WBPH emigrates from the northern part of southern China starting in late May, approximately one month earlier than BPH [21]. Thus, questions arise as to whether the northward migration of WBPHs in eastern China was similarly affected by the WPSH and which weather system plays a key role in WBPH migration, if not the WPSH. This paper presents long-term WBPH light traps (1977–2003, the same years as the previous work on BPH migration) and meteorological data to investigate this issue. We investigate links between seasonal weather systems and WBPH migration patterns that are relevant for forecasting migration events. Such information is necessary for migratory pest control.

## 2. Materials and Methods

### 2.1. Light-Trap Data

To compare with the previous study, [8], on the BPH migration pattern linked to the WPSH, a similar 27 year dataset of daily WBPH light-trap data (1977–2003) was used in this study (Figure 1). This dataset was collected from standardized 20 W blacklight (UV) traps located at plant protection stations (PPS) of 222 counties in China. Furthermore, data from 20 stations, that had complete data coverage from 1978 to 2003, were used for further detailed analysis (Figure 1).

### 2.2. WBPH Migration Volume and Concentration Zone

The southward migration was not studied in this study. Therefore, the light-trap data from 1 April–30 August of each year in 1977–2003 were selected. To represent WBPH migration activity, the total catch for each five-day period for each station were summed. Then, we calculated the 90th percentile value of catches (termed ‘*WBPH90th*’) in each of the 810 five-day periods from 1 April–30 August (the 30 5-day periods × 27 years). Finally, the seasonal variation in WBPH immigration volume was presented by using the mean value of 27 years for each 5-day period (Figure 2a).

To explore the seasonal variation in the location of WBPH concentration zones (i.e., a station that experienced a major WBPH immigration), the latitude-time cross-section of the relative 2-D binned kernel density of stations in a concentration zone was estimated. First, any given WBPH trapping station in any 5-day period was defined as a planthopper ‘concentration zone’ if the number of WBPH in the 5-day catch was greater than or equal to the WBPH90th (i.e., the 90th percentile value in that period and that year). A total of 10,978 records of stations in a ‘concentration zone’ in all 810 periods (the 30 5-day periods × 27 years) were extracted from all records of 222 stations from 1977–2003. Second, based on these 10,978 records, the latitude-time cross-section of the 2-D binned kernel density of stations in the concentration zone was estimated by using the ‘*bkde2D*’ function (http://stat.ethz.ch/R-manual/R-devel/library/KernSmooth/ html/bkde2D.html) in R software (version 3.4.1, https://www.r-project.org/).

### 2.3. Meteorological Data and WPSH Indices

All global-gridded meteorological data with a spatial resolution of 2.5° (i.e., 144 points in longitude and 72 points in latitude) were downloaded from National Oceanic and Atmospheric Administration’s (NOAA) Earth System Research Laboratory (ESRL, http://www.esrl.noaa.gov/). Among these data, the Climate Prediction Center Merged Analysis of Precipitation (CMAP) data include monthly and five-day precipitation means since 1979. The monthly and daily geopotential height, u-winds, and v-winds were derived from the National Centers for Environmental Prediction (NCEP)/National Center for Atmospheric Research (NCAR) reanalysis data from 1948 to 2011.

The monthly indices of WPSH (110° E to 180° E) from 1951 to 2010 were downloaded from the China Meteorological Data Sharing Service System (CMDSSS, http://cdc.cma.gov.cn/). There are five WPSH indices used to describe the area, intensity, mean ridge, westward extension, and north edge.

The five-day mean ridge of the WPSH was calculated using daily NCEP/NCAR reanalysis data. The WPSH was described by the region of 5860 gpm at 500 hPa. The area and the mean geopotential height of this region are defined as the area and intensity indices of WPSH. The mean latitudinal position of the WPSH ridge, the longitude of the westernmost point, and the latitude of the northernmost point were defined as the mean ridge, the westward extension, and the north edge indices of the WPSH, respectively. The ridge of the WPSH is also defined as the boundary between the east wind and the west wind (WEB). Thus, the five-day latitude location of the WPSH ridge was derived from the WEB, which was calculated by the following equation [8,17,18,22]:(1)$$u=0;\frac{\partial y}{\partial x}=0$$
where u is the speed of the zonal wind and coordinate (x, y) is its location in the 2-D dimension of u-wind grid data. The location of the five-day mean ridge was calculated by the Grid Analysis and Display System (GrADS, version 2.0.2, http://grads.iges.org/grads/).

### 2.4. Spatial Exploration of Correlations Between WBPH Catches and WPSH Indices, Precipitation

To explore the spatial distribution of WBPH immigration in China, we interpolated the raster surfaces of WBPH catches from the PPS locations using the natural neighbor method in ArcGIS software (version 10.2, http://www.esri.com/). We tested for correlations between WBPH catches and WPSH indices at 20 PPS using Pearson correlations. Results of the correlations were visualized using R software.

Four representative sites with interannually complete data from 1977–2003, including Qujiang, Quanzhou, Yongfu, and Ziyuan (the red triangles in the green oval area in Figure 1), and the WBPH catches from these four sites were summed as the average volume of WBPH migration in the northern part of southern China during 16 May–15 June each year. The precipitation of this period in each year was derived from five-day CMAP data. A spatial map of correlation coefficients between the immigration levels in the northern part of southern China and the precipitation from 1979–2003 was calculated and visualized by using GrADS.

### 2.5. Regression Models of WBPH Immigration Volume Against Distance

We explored the relationship between the WBPH immigration volume at each station and the migration distance from southern China using linear models. The WBPH immigration volume at a station was defined as the total number of light-trap catches during a month long period, 16 May to 15 June for Period I or 16 June to 15 July for Period II. These two periods correspond to two major emigration waves (see Results section). These immigration volumes were log transformed with a log base of 10 to follow a normal or near normal distribution before modeling. Yongfu station, in the northern part of southern China, was defined as the hypothetical starting point. The stations located northeast of Yongfu were selected as the WBPH landing area (blue dots in Figure 1). The migration distances from Yongfu to each site were calculated using the following equation:(2)C=sin(LatA)×sin(LatB)×cos(LatA−LonB)+cos(LatA)×cos(LatB)
(3)Distance=R×Arccos(C)×π/180
where (LonA, LatA) represent the longitude and latitude of the first site A (i.e., Yongfu), (LonB, LatB) represent the longitude and latitude of another site B, and R is the radius of the earth.

## 3. Results

### 3.1. Association of the WPSH System with the Northward Migration of the WBPH

The seasonal expansion of WBPHs in eastern China is schematized in Figure 2a,b. Just before June, the migration intensity increased over time (Figure 2a), but the concentration zone remained in southern China before mid-June (Figure 2b). This indicated that southern China was the area most seriously affected by the WBPH. The migration occurred after mid-June and the concentration zone shifted toward the Lower Yangtze River Valley (Figure 2c and Figure 3). This step synchronized with the following events: (i) The southwesterlies expanded to the north and the winds strengthened in eastern China (Figure 2c), (ii) the rainfall belt moved to the Lower Yangtze River Valley (Figure 2d), and (iii) the ‘first abrupt jump’ of the WPSH occurred (Figure 2a). Here, the first two events, the movement of the rainfall belt (a barrier to RPH migration) and the expansion of the southwesterlies (rapid aerial transport), would directly affect WBPH migration. However, these two events were determined or indicated by the third event, the ‘first abrupt jump’ of the WPSH, expanding abruptly northward from southern China to the Yangtze River Valley, heralding the Meiyu season in the latter region. In summary, the wave of WBPH migrants into the Lower Yangtze River Valley depends on the development of the WPSH, which regulates southwesterly airstreams and the location of rain belts. For more detail, Figure 3 shows that the northward shift of the WBPH concentration zone coincided with the expansion of the WPSH. The concentration zone was located north of the WPSH ridge close to the 5860 gpm contour at 500 hPa altitude, whereas the WPSH ridge itself was located south of 30° N (Figure 3). As the WPSH moved northward in June and July, the WBPH concentration zone showed a corresponding northward shift (Figure 3).

From monthly correlation coefficients between WBPH cumulative light-trap catches at the 20 PPS stations and WPSH indices (Figure 4), it was shown that WBPH catches in PPS sites in the Yangtze Valley (Jianli, Huizhou and Dongzhi) were significantly correlated with WPSH intensity. This result also indicated that the WBPH migration volume in the Yangtze River Valley was influenced by the WPSH intensity in June. In addition, there is another cluster of grids in Figure 4 with significant correlations in the intensity index panel, i.e., the WBPH catches in PPS sites in the northern part of southern China (Zhaoqing, Shantou, Quanzhou, Ziyuan, Qujiang, and Fuqing) were correlated significantly with WPSH in August, albeit negatively. This cluster of grids was not considered in this study. Due to the significant correlation between the intensity and area indices of WPSH (Pearson’s correlation coefficient *r* = 0.93, *p* < 0.0001, in 1977–2016) [9], some similar patterns of grid clusters also appeared in the area index panel, but these patterns were less significant (Figure 4).

### 3.2. Short Migration Distance of the WBPH in Late May and Early June

A previous study suggested that the WBPH emigrates from the northern part of southern China starting in late May [21], but the concentration zone of WBPHs shifted to the Lower Yangtze River until the ‘first abrupt jump’ of the WPSH happened in mid-June (see the previous section). Therefore, we assumed that the emigration of WBPHs in late May and early June (i.e., the fourth migration step, see Figure 2a and Figure 3) was unsuccessful or of a short distance. The migration distance of the WBPH in two periods (Period I: 16 May to 15 June; Period II: 16 June to 15 July) was compared by analyzing the number of light-trap catches at each station along its migration pathway. Yongfu station, in the northern part of southern China, was used as the starting point, and the number of light-trap catches decreased as the linear distance from the starting point (i.e., Yongfu) increased in both periods. The decrease rate in Period I was much faster than that in Period II (Figure 5). The absolute value of the slope in the fitted linear model for Period I (slope value = −0.15, Standard Error (S.E.) = 0.009, t = 16.5, *p* < 0.0001) was larger than that for Period II (slope value = −0.02, S.E. = 0.006, t = 3.265, *p* = 0.0011) (Figure 5). This result indicated that the migration distance of the WBPH in Period I was significantly shorter than that in Period II. The spatial distribution of WBPH immigration in Period I also showed that most WBPH catches were found in the northern part of southern China due to their short-distance migration (Figure 2c, Figure 3, and Figure 5).

### 3.3. WBPH Migration is Hindered in the Preflood Season from May to June in the Northern Part of Southern China

The rain belt mostly stayed in southern China in May and June before it shifted to the Yangtze River Valley, and this period is locally named the preflood season. Hence, we assumed that the WBPH emigration was hindered by this continuous large-scale rainfall in southern China, and, thus, the correlation between them was tested. WBPH from four representative sites (Qujiang, Quanzhou, Yongfu, and Ziyuan) were summed as the average volume of WBPH migration in the northern part of southern China during 16 May–15 June each year. A spatial map of correlation coefficients between the immigration levels in the northern part of southern China and the precipitation from 1979–2003 revealed a significant correlation between catches and rainfall in the region immediately to the north (see the red region in Figure 6). These four stations are located at the southern fringe of the rain belt, which thus forms a natural barrier to migration at this time (Figure 2d and Figure 6).

## 4. Discussion

To explore the link between insect migration and seasonal weather systems, the northward migration pattern of the WBPH in eastern China was described in this study. It revealed that the WBPH migrated from southern China into the Yangtze River Valley after ‘the first abrupt jump’ of the WPSH in mid-June. This result is consistent with our previous studies on BPH migration [8,9]. Therefore, this study presented another case for how the seasonal WPSH weather system shapes the route and timing of insect migration. The seasonal weather system can influence insect migration, largely because it establishes the prevailing winds and seasonal rainfall over thousands of square kilometers or even large portions of the earth for days or weeks. Here, after the ‘first abrupt jump’ of WPSH in mid-June, the wide zonal rainfall (a barrier forcing RPH landings) moved to the Yangtze River Valley, while the southwesterly airstream (a high-speed vehicle for RPH migration) strengthened and expanded to the Yangtze River Valley [8,9]. In this study, the WBPH could not migrate further north before mid-June, mostly because its emigration in southern China was hindered by heavy rainfall, which also signified that the seasonal weather system and its associated weather conditions are vital for insect migration.

Although both the BPH and the WBPH migrate into the Yangtze River Valley after mid-June, the timings of their immigration peaks in the Yangtze River Valley are different. In mid- and late June, most BPH migrants came from the southern part of southern China (south of the Tropic of Cancer, ~ 23.5°N), and its concentration zones were located in the northern part of southern China and the Yangtze River Valley. The number of migrants decreased as the migration distance increased, so BPH catches in the Yangtze River Valley were much smaller than those in the northern part of southern China [8,14]. In contrast, this study showed that WBPH migrants during these times were from the northern part of southern China. The migration distance to the Yangtze River Valley for the WBPH is much shorter than that of the BPH from the southern part of southern China, and thus, a larger percentage of WBPH individuals, compared to BPH individuals, were able to reach the Yangtze River Valley. Due to interspecific competitive disadvantage [23,24,25,26], the WBPH population in southern China was soon replaced by the BPH population, and a few WBPH migrated into the Yangtze River in July. The immigration peaks of the WBPH in the Yangtze River Valley occurred only in late June and early July. In contrast, the immigration peaks of the BPH in the Yangtze River did not occur until most arrivals had migrated from the northern part of southern China in July [8,9,27]. Therefore, the population dynamics of these two RPH species were still different in the Yangtze River Valley. The WBPH population settled earlier than the BPH population [26,27,28,29]. However, the WBPH is at a competitive disadvantage when both RPH species are established in the same habitat [23,24,25,30]; therefore, the growth of the WBPH population did not last long in the Yangtze River Valley (Figure 2a), and WBPH populations decreased and were replaced by BPH [31]. As a consequence of this, the WBPH was not a serious threat to rice crops in this region and was not paid much attention by farmers, agricultural managers, or scientists there [31].

In southern China, farmers practice a double-cropping rice system and the early season rice is planted in late March or early April. At the same time, the WBPH migrates into this area from countries to the south, such as Vietnam, Laos, and Thailand [16,19,32,33]. More specifically, the first migrants arrive in the southern part of southern China in late March [16,34] and the northern part of southern China in mid- and late April [21,35]. However, BPHs arrive at the above regions approximately one month later [14,16,27]. There is a lack of competitors and natural enemies, and the verdant and juicy early season rice plants at the tillering stage are suitable for the WBPH, so the WBPH population grows very fast [32]. In this study, it was shown that the WBPH emigration was unsuccessful or only covered a short distance due to the heavy rainfall during the preflood season. Finally, the maximum number of WBPH light-trap catches occurred in southern China in early June (Figure 2a). Therefore, the occurrence of the WBPH was more serious than that of the BPH [32].

## 5. Conclusions

In this study, the migration pattern of the WBPH was described. Due to the similar body sizes of the WBPH and the BPH, both their migration patterns are influenced by the WPSH. Both RPHs in southern China (south of approximately 27° N) migrates into the middle and lower reaches of the Yangtze River after the abrupt movements of the WPSH in mid-June. Nonetheless, considering the differences in their interspecific competition ability and life cycle, these two species had different population dynamics in different regions. The migration of the WBPH in late May and early June was short or unsuccessful due to heavy precipitation during the preflood season in southern China from late May to middle June, resulting that the threat of WBPHs to rice was more serious than that of BPHs in southern China. Nevertheless, our study improves our understanding of the relationship between insect migration and seasonal atmospheric circulation.

## Figures and Tables

**Figure 1 insects-10-00158-f001:**
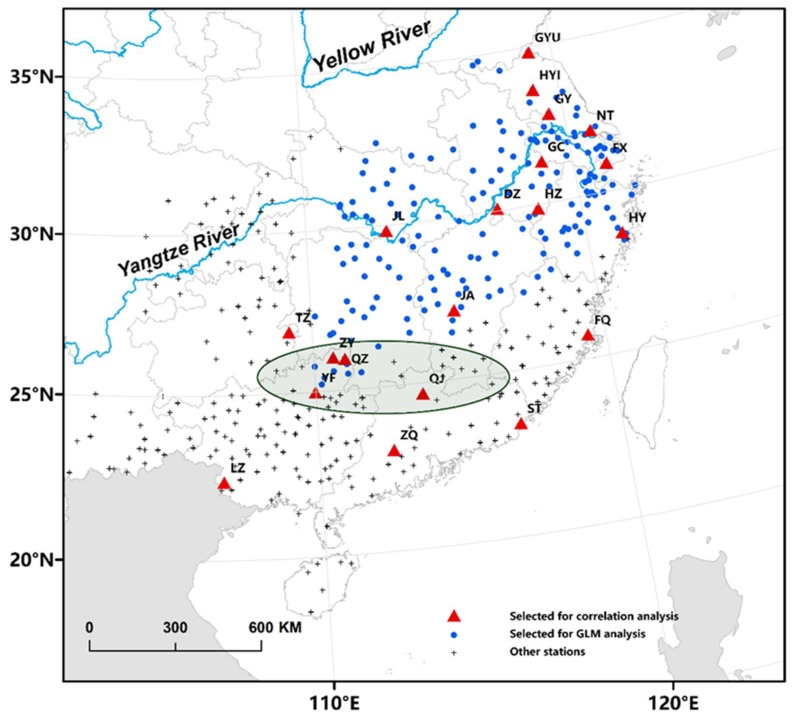
Locations of 222 county plant protection stations (PPS) with white-backed planthopper (WBPH) data (black cross symbol). The representative stations, with complete data from 1978–2003, were selected for further detailed analysis and are represented by red triangles, namely, Longzhou (LZ), Zhaoqing (ZQ), Shantou (ST), Qujiang (QJ), Quanzhou (QZ), Yongfu (YF), Ziyuan (ZY), Tianzhu (TZ), Ji’an (JA), Fuqing (FQ), Jianli (JL), Dongzhi (DZ), Huizhou (HZ), Huangyan (HY), Gaochun (GC), Fengxian (FX), Nantong (NT), Gaoyou (GY), Huaiyin (HYI), and Ganyu (GYU). From 16 May to 15 July, most WBPHs come from the northern part of southern China (approximately 23.5–27° N, green oval area), and we selected Yongfu as a hypothetical starting point. The stations located northeast of Yongfu were selected as the WBPH landing areas (blue dots).

**Figure 2 insects-10-00158-f002:**
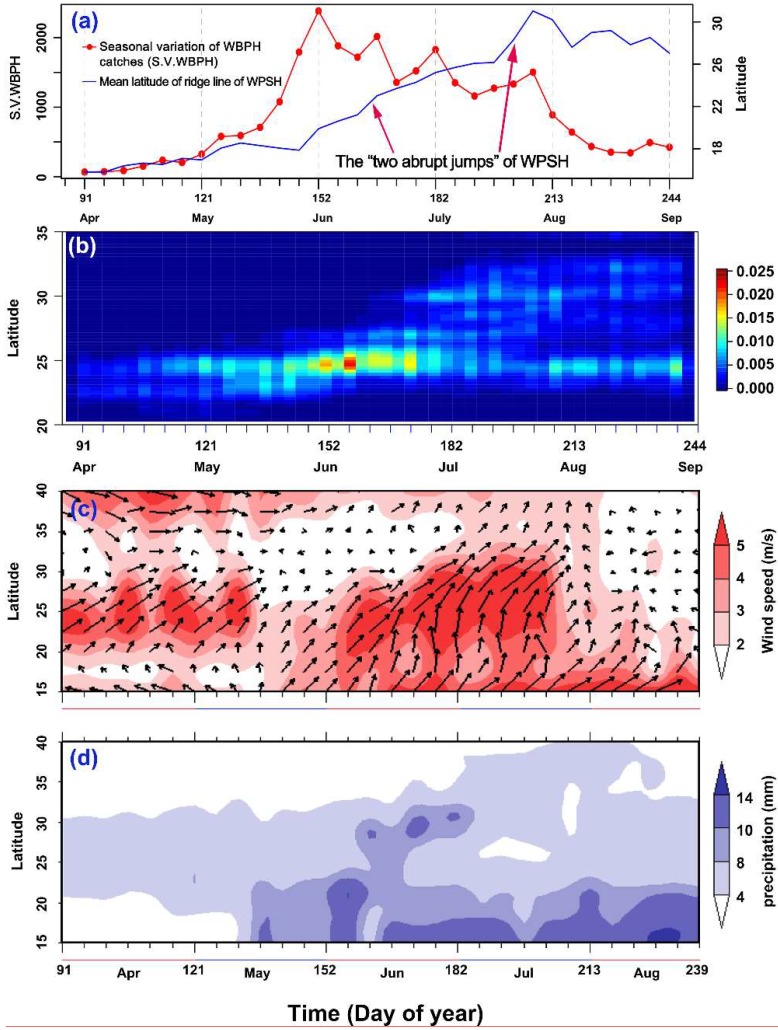
(**a**) The mean latitude of the WPSH ridge between 110° E and 120° E and the mean seasonal variation in 5-day WBPH catches in eastern China. The ‘two abrupt jumps’ of the WPSH are meteorologically defined (e.g., Tao & Wei 2006; Ding et al. 2007) [17,18] and denote the beginning and end of the Meiyu season in the Yangtze River Valley. (**b**) The location of the WBPH concentration zone in 5-day periods in eastern China, represented by the relative density of traps per unit of latitude for each 5-day period in a WBPH concentration zone. Any given WBPH trapping station in any 5-day period was defined as a planthopper ‘concentration zone’ if the number of WBPHs in the 5-day catch was greater than or equal to the WBPH90th (i.e., the 90th percentile value in that period of that year. (**c**) 27 year (1977–2003) mean winds at 850 hPa height in 5-day periods between 110° E and 120° E. (**d**) 27 year (1977–2003) mean precipitation in 5-day periods between 110° E and 120° E.

**Figure 3 insects-10-00158-f003:**
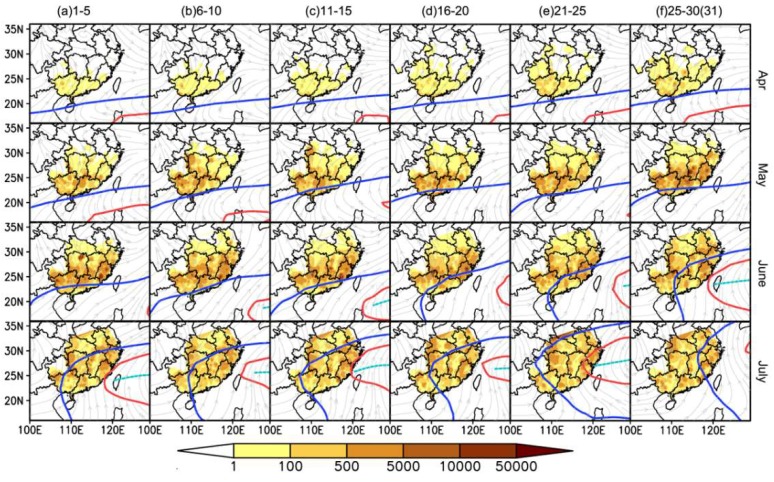
The spatial distribution of WBPH immigration intensity (yellow to brown areas), based on mean catches from 1977 to 2003 and the mean WPSH range for five-day periods from April to July. The raster surfaces of WBPH catches from 222 county PPS sites were interpolated using the natural neighbor method in ArcGIS. The range of the WPSH is shown using a 500 hPa geopotential height contour. The red solid lines show the contour at 5880 gpm, the blue solid lines show the contour at 5860 gpm, and the cyan dashed lines show the ridge line of the WPSH.

**Figure 4 insects-10-00158-f004:**
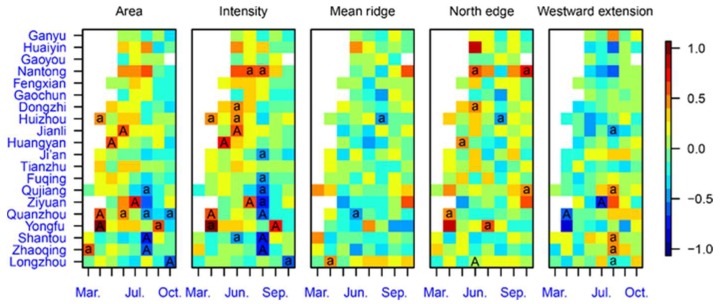
Monthly correlation coefficients between cumulative light-trap catches of WBPHs (after log transformation) at the 20 PPS sites and WPSH indices. Sites are listed geographically, from north to south, and positive and negative correlations significant at the 1% and 5% levels are labeled with ‘A’ and ‘a’, respectively.

**Figure 5 insects-10-00158-f005:**
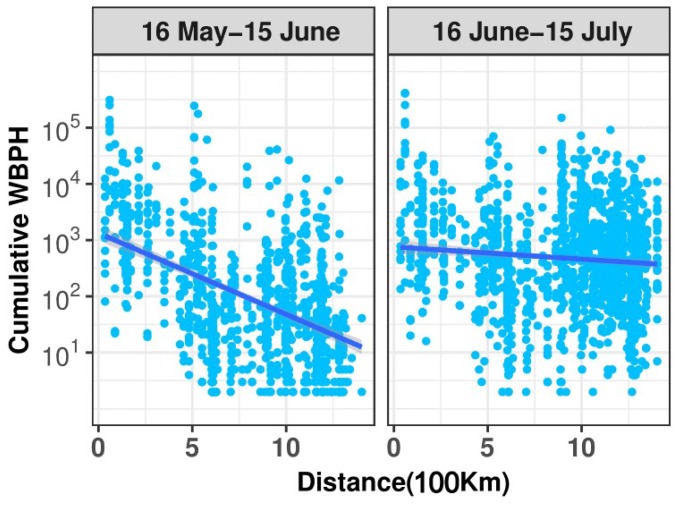
The relationship between the WBPH immigration volume and migration distance. Yongfu station in Guangxi was used as the starting point. PPS sites located northeast of Yongfu were selected and the distances from Yongfu to each selected station were calculated.

**Figure 6 insects-10-00158-f006:**
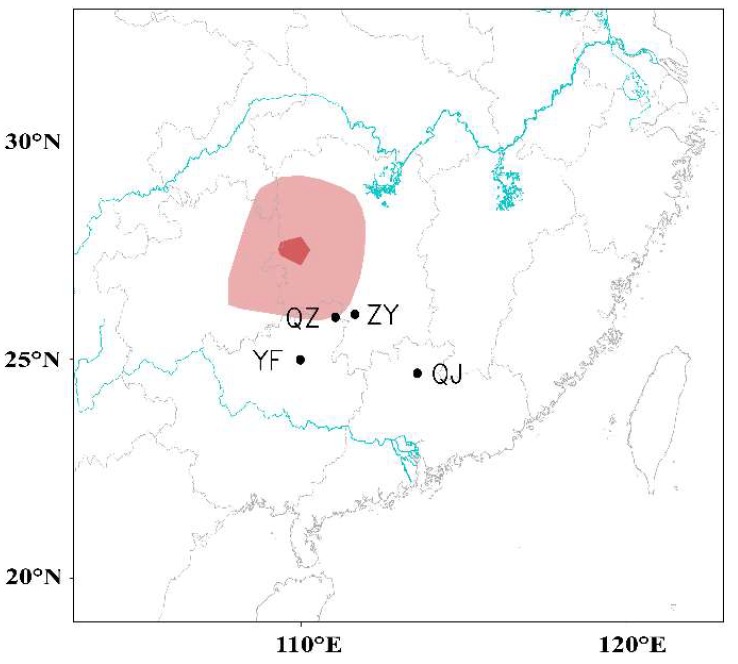
Simultaneous correlation map between WBPH immigration volume in the northern part of southern China and precipitation during 16 May–15 June. The light-trap catches from four PPSs, namely, Yongfu (YF), Quanzhou (QZ), Ziyuan (ZY), and Qujiang (QJ), during 16 May–15 June were summed each year as the WBPH immigration volumes in 1978–2003. The light and dark red areas indicate significance at the 1% and 5% levels, respectively.

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
