# Peer review of "The Early Northward Migration of the White-Backed Planthopper (Sogatella furcifera) is Often Hindered by Heavy Precipitation in Southern China during the Preflood Season in May and June"

_insects, 2019, doi:10.3390/insects10060158_

Round 1

Reviewer 1 Report

For insect migration, it is a common knowledge that strong downdraft wind or precipitation would force any airborne migrating insects to land. Therefore, it is expected that in the typical monsoon arena of eastern China immigration would occur in the rain zone (northwest edge of the western pacific subtropical high pressure). However, it is still worthwhile to demonstrate the evidence that this is the case of the early immigration of white backed planthopper into the Lower Yantze River.

Things could be improved:

1) Introduction: moving in "WPSH system annual jumps" from Results - it's other people's finding. Also it would be good to explain this as the WBPH migration background.

2) Materials and Methods:

a) WBPH immigration volume and immigration center: simply based on light-trap data and 2D binned kernel density can't determine "concentration and landing zone". It is sufficient to say it is adult abundant location but not necessary "immigration event."

b) Regression models of WBPH immigration volume against distance: you have not shown Yongfu is the dominant emigration center during the study period, and thus regression against it is fundamentally biased. For example, the statistics of short-distance migrations could simply be some emigration places further way from Yongfu.

In addition, for the distance calculation, it is not necessary to get to that kind of precision (as other data don't have very high resolution). Thus, a simpler triagonometric estimate with 100km per longitude degree and 110km per latitude degree would be enough, as we are not dealing with high latitudes.

3) Results: it should only show what you have found in this research, nothing from other people or previous studies - they belong to either Introduction or Discussion.

In addition, should the title "The early(initial) northward migration of white backed planthopper (Sogatella furcifera) is often hindered by heavy precipitation in the southern China in May and June" be better to state the study?

"pre-flood season" is not a well-defined term for key word.

As I said in the beginning, this is a topic worthwhile to study, but I'm afraid the draft is not up to the publication standard yet.

Author Response

Thanks very much.

1) Introduction: moving in "WPSH system annual jumps" from Results - it's other people's finding. Also it would be good to explain this as the WBPH migration background.

 3) Results: it should only show what you have found in this research, nothing from other people or previous studies - they belong to either Introduction or Discussion.

>>>  I moved / deleted most other people’s findings in Results section. But I still reserved some sentences of this kind to make the story more smoothly. “WPSH system annual jumps” in Results was moved into Introduction Part, and ‘the preflood season’ was also described. 

2) Materials and Methods:

a) WBPH immigration volume and immigration center:

>>> Title ‘WBPH immigration volume and immigration center’ was changed to ‘WBPH migration volume and concentration zone’. The term ‘"concentration and landing zone’ was changed to ‘"concentration zone’.

b) Regression models of WBPH immigration volume against distance:

>>>  Previous study suggested most migrating WBPH in May and June was come from the northern part of southern China.  We did not have evidence to say that Yongfu just was the dominant emigration center during this study period, but we just want a hypothetical starting points to calculate the distance, and any hypothetical starting in southern China will not have effect on the distribution of light trap catches along the migration route. As you said it is not necessary to get to that kind of precision.

In addition, should the title "The early(initial) northward migration of white backed planthopper (Sogatella furcifera) is often hindered by heavy precipitation in the southern China in May and June" be better to state the study?

>>>  Changed to “The early northward migration of the white-backed planthopper (Sogatella furcifera) is often hindered by heavy precipitation in southern China during the preflood season in May and June”

"pre-flood season" is not a well-defined term for key word.

>>> Removed from key word list.

Reviewer 2 Report

This manuscript analyzed the relationship between northward summer migration of the white-backed planthopper, Sogatella furcifera in China and a synoptic weather system. It found that S. furcifera emigrates from southern China in mid-May to mid-June, about one month earlier than the timing of major emigration of the brown planthopper, and that these emigrations are short-distance migrations with landing area located in a northern part of southern China. This short migration occurs because the Meiyu front is located over there, being affected by the location of a summer high pressure system on western Pacific Ocean. Therefore, the timing and landing areas of the summer migration of the two major rice planthoppers are different in China.

Following are minor comments. Especially, Materials and Method section can be improved. English writing seems premature and should be improved.

The leading number is a line number of the manuscript.

93: 20 stations. Is this right? In Fig. 4, 18 stations are listed.

Figure 1: Show the location of 18 stations in Fig. 4, please. And an explanation of blue dots should be added in the caption.

143: Is the ridge of WPSH defined by the boundary between easterly wind and westerly wind? If so, that definition is added.

159: How did you select blue-dot stations? Were they selected based only on their location?

160: landing area à landing area (blue dots in Fig. 1)

194, Fig.4 and Figure 6: Describe the methods to make these figures, which is not seen in Materials and Methods section. For Fig. 4, the definition of 5 parameters of WPSH.

239: An expression that WBPH is hindered by season is odd to me.

250: (Figs. 2e, 6) à (Figs. 2d, 6)?

“South China” is often used. Foreign readers do not know exactly where it is. Describe that area at first appearance point of the phrase.

English is odd some time. For example, the South China should be South China, "Brown planthopper" in line 17 should be "the brown planthopper", "the migration pattern for WBPH" in line 21 should be "migration patter of WBPH", "of the WPSH" should be "of WPSH", and so on. There are some more which are not described here. Please have a native proofreading.

Author Response

Thanks very much.

93: 20 stations. Is this right? In Fig. 4, 18 stations are listed.

>>>  20 stations is right. The Fig. 4 is updated with new 20 stations’ data.

Figure 1: Show the location of 18 stations in Fig. 4, please. And an explanation of blue dots should be added in the caption.

>>> Fig. 1 was updated.  Now there are 419 stations for monitoring rice planthoppers in China, but only the data from 1977-2003 were chose and analyzed to compare with the previous study on the BPH migration. In these time period, there were only 222 stations, thus new Fig. 1 only showed these 222 stations (but old one showed all 419 stations.)

143: Is the ridge of WPSH defined by the boundary between easterly wind and westerly wind? If so, that definition is added.

>>> The definitions of all five WPSH indices were added.

159: How did you select blue-dot stations? Were they selected based only on their location?

>>> Just based on their location, all these site were located to the northeast of the Yongfu station. Fig. 1 was updated. The sites didn’t work in 1977-2003 were deleted.

160: landing area à landing area (blue dots in Fig. 1)

>>> Changed.

194, Fig.4 and Figure 6: Describe the methods to make these figures, which is not seen in Materials and Methods section. For Fig. 4, the definition of 5 parameters of WPSH.

>>> A section titled ‘Spatial exploration of correlations between WBPH catches and WPSH indices, precipitation’ was added in ‘Material and method’  

239: An expression that WBPH is hindered by season is odd to me.

>>> Changed into ‘WBPH migration is hindered in the preflood season from May to June in the northern part of southern China’

250: (Figs. 2e, 6) à (Figs. 2d, 6)?

>>> Changed.

South China” is often used. Foreign readers do not know exactly where it is. Describe that area at first appearance point of the phrase.

>>> All ‘South China’ was replaced by southern China, and southern China was defined as the south of approximately 27°N.

English is odd some time.

>>> English writing was improved by Willey Editing Serves (http://wileyeditingservices.com/en).

Round 2

Reviewer 1 Report

This draft is much better than the previous one, but still have room to improve. I list my suggestions/comments here with line numbers:

15: determine/establish

17: results/studies

24: determined/suggested

25: migrated/migrates

26: However/By contrast

27: began/begins

27: in late May. => in late May, which is earlier than that of BPH. (Is this correct?)

29: demonstrate/demonstrated

30-31: new information: what is this "new information"? Say it here.

33: replace "Nilarparvata lugens" with WPSH

37: every year[cite Hu's Science paper here].

41: remove annual; journey distances / journeys; hundreds of/hundreds

42: stop flying/terminate their migration

44: concentrated on the ground in either favorable or unfavorable conditions/landing in concentration density

46: contending: insects not necessarily contend with weather conditions, but most likely to choose the right wind to ride on.

48: systems/patterns; determine/establish

50: determine/control

51-52: remove terminate insect migration

55: significance/importance

58: In outbreak years, this / Outbreaks

68: north, and ...=> north, which produces several continuous precipitation events in southern China in April and May.

69: During the summer season each year => During summer season

70: abruptly splits into two northerly movements => hops abruptly northwards in two steps.

84-89: duplicated, should be reworded.  In addition, delete the last sentence of 86-87.

93-95: move this into Discussion

99: We identify/We investigate

105: move citation [8] to 104 right after "previous study"

110: reorganize this sentence by: The light-trap data from ..

122: From 16 May ... as another paragraph, not included in the caption.

187-188: this sentence should be in Results.

197-198: as I addressed before, this is unnecessary fine resolution, as your weather data is 2.5 degree grids, which are roughly at 250km resolution.

247-248: Move into Discussion

250: hypothesized: normally it should be followed by a form statistical test, but not the case here. So I suggest you use "assumed" or similar word instead.

269: "hypothesized": see above

291: determines/establishs

331: "more serious": explain it to make it clear

Although this revision makes a lot of senses, it should be read thoroughly to avoid any duplicates, and mixed topics in one paragraph. The BIG impression is there are too many "however" - check what you try to say and use other link words.

Author Response

51-52: remove terminate insect migration

>> The sentence will not be completed if delete this. Wind pattern shape migration pathway, rain terminate insect migration.

197-198: as I addressed before, this is unnecessary fine resolution, as your weather data is 2.5 degree grids, which are roughly at 250km resolution.

>> Agree. But we still need a method to calculate the distance.